# Autologous Platelet Concentrates in Treatment of Furcation Defects—A Systematic Review and Meta-Analysis

**DOI:** 10.3390/ijms20061347

**Published:** 2019-03-17

**Authors:** Sourav Panda, Lorena Karanxha, Funda Goker, Anurag Satpathy, Silvio Taschieri, Luca Francetti, Abhaya Chandra Das, Manoj Kumar, Sital Panda, Massimo Del Fabbro

**Affiliations:** 1Department of Biomedical, Surgical and Dental Sciences, Università degli Studi di Milano, 20122 Milan, Italy; drsaurav87@gmail.com (S.P.); lorikaranxha@gmail.com (L.K.); fundagoker@yahoo.com (F.G.); silviotaschieri@gmail.com (S.T.); luca.francetti@unimi.it (L.F.); 2Institute of Dental Science and SUM Hospital, Siksha O Anusandhan University, Bhubaneswar 751003, India; dearanurag@gmail.com (A.S.); drabhaya2011@gmail.com (A.C.D.); manojkumar@soa.ac.in (M.K.); 3Dental Clinic, IRCCS Istituto Ortopedico Galeazzi, 20161 Milan, Italy; 4Department of Public Health, Regional Medical Research Center, Bhubaneswar 751003, India; drsheetalpanda@gmail.com

**Keywords:** autologous platelet concentrates, bone defects, bone grafting, bone regeneration, furcation defects, periodontal defects, periodontal regeneration, periodontal surgery, platelet-rich plasma, platelet-rich fibrin, plasma rich in growth factors, tissue healing

## Abstract

Background: The aim of this review was to evaluate the adjunctive effect of autologous platelet concentrates (APCs) for the treatment of furcation defects, in terms of scientific quality of the clinical trials and regeneration parameters assessment. Methods: A systematic search was carried out in the electronic databases MEDLINE, SCOPUS, CENTRAL (Cochrane Central Register of Controlled Trials), and EMBASE, together with hand searching of relevant journals. Two independent reviewers screened the articles yielded in the initial search and retrieved the full-text version of potentially eligible studies. Relevant data and outcomes were extracted from the included studies. Risk of bias assessment was also carried out. The outcome variables, relative to baseline and post-operative defect characteristics (probing pocket depth (PPD), horizontal and vertical clinical attachment loss (HCAL, VCAL), horizontal and vertical furcation depth (HFD, VFD) were considered for meta-analysis. Results: Ten randomized trials were included in this review. Only one study was judged at high risk of bias, while seven had a low risk, testifying to the good level of the evidence of this review. The meta-analysis showed a favorable effect regarding all outcome variables, for APCs used in adjunct to open flap debridement (*p* < 0.001). Regarding APCs in adjunct to bone grafting, a significant advantage was found only for HCAL (*p* < 0.001, mean difference 0.74, 95% CI 0.54, 0.94). The sub-group analysis showed that both platelet-rich fibrin and platelet-rich plasma in adjunct with open flap debridement, yielded significantly favorable results. No meta-analysis was performed for APCs in combination with guided tissue regeneration (GTR) as only one study was found. Conclusion: For the treatment of furcation defects APCs may be beneficial as an adjunct to open flap debridement alone and bone grafting, while limited evidence of an effect of APCs when used in combination with GTR was found.

## 1. Introduction

Furcation involvement is defined as bone resorption and attachment loss in the inter-radicular space that results from plaque associated periodontal disease [1]. The treatment of periodontal disease associated with furcation represents a challenge for the clinician, due to the complexity in anatomy and morphology of such area. The unfavorable anatomic feature of the furcation restricts adequate instrumentation for proper debridement, thereby limiting the prognosis of the involved teeth [2].

Various treatment modalities, including surgical and non-surgical therapy, have been proposed to improve the prognosis based on the degree of furcation involvement. Several classifications have been proposed over the years (Table 1), based either on the severity of horizontal probing depth into the furcation defect or on the vertical amount of alveolar bone loss within the defect [3]. The most popular one was developed by Glickman, which divides furcation defects into four grades [4]. Non-surgical strategies such as scaling and root planing, furcation-plasty, etc. are employed to treat the furcations with Grade I initial involvement which restores the gingival health. Conversely, surgical procedures including regenerative and resective approaches, are performed for the treatment of more advanced lesions, to allow access to the internal complex areas of furcations. The traditional resective approach may negatively affect the long term prognosis of the treated teeth, however, it is considered as the treatment of choice for grade III and IV furcation lesions, aiming at facilitating maintenance of the furcation area.

Regenerative approaches are aimed at furcation closure by the formation of new bone, cementum and periodontal ligament in the involved inter-radicular space. Thorough debridement with adequate instrumentation following surgical exposure of furcation involved area, is one of the earliest and most well-documented treatment protocols to achieve regeneration in grade II furcation lesions [5]. In addition, various studies were carried out in the recent past, using bone substitutes, barrier membrane, autologous, and recombinant growth factors in order to provide evidence of improved bone fill and attachment gain in treating grade II furcation lesions [6,7].

The use of biologic agents consisting of growth and differentiation factors like rhBMP2 (recombinant human bone morphogenetic protein-2), rhPDGF (recombinant human platelet-derived growth factor), and TGF-β (transforming growth factor beta), had proven to promote osteogenic induction in cases of furcation treatment, in animal studies [8,9,10]. Additionally, the use of autologous platelet concentrates (APCs) is gaining popularity as a source of multiple growth factors in high concentrations, for regenerative treatments in many clinical applications. The contribution of blood-derived platelets to the bone healing process is mainly based on the growth factors stored in their granules and released upon activation. Autologous platelet concentrates are advantageously used as a cost-effective adjunct to surgical regenerative therapy, even in combination with bone grafts and barrier membranes. Several randomized controlled trials have reported on the efficacy of the use of these APCs when used alone or in combination with various regenerative strategies and other biologic agents, suggesting improvement of post-operative soft and hard tissue healing, and improved bone fill and attachment gain [6,11,12,13,14,15,16,17]. Different types of APCs are available, each with peculiar features, the most popular being platelet-rich plasma (PRP), along with platelet-rich fibrin (PRF), plasma rich in growth factors (PRGF) and concentrated growth factors (CGF).

A recent systematic review and meta-analysis [18], evaluating the effect of use of platelet-rich fibrin (PRF) in adjunct to open flap debridement, included two clinical trials in treatment of grade II furcation with nine month follow-up and concluded favorable results with the use of PRF in terms of clinical attachment level gain (mean difference 1.25 CI 95% [0.82, 1.65], *p* = 0.07) and bone fill (mean difference 1.52 CI 95% [1.18, 1.87], *p* = 0.05). Another systematic review [19] included two split-mouth clinical trials evaluating the effect of platelet-rich plasma and reported no consistent evidence regarding the effect of PRP in treatment of furcation defects.

The aim of this systematic review is to evaluate the adjunctive effect of APCs in treatment of furcation defects both qualitatively and quantitatively, in terms of scientific quality of the clinical trials and regeneration parameters assessment.

## 2. Results

### 2.1. Study Characteristics

A total of ten studies were included in this systematic review after independent screening of titles and abstracts from a pool of 153 articles retrieved from the search platforms. The systematic flow chart of the study selection process is provided in Figure 1. Out of 21 eligible studies, 11 studies were excluded with reasons provided in Table 2. The general information and the study characteristics of the included studies are detailed in Table 3.

The general comparison was between a group that received APC as an adjunct to surgical treatment (experimental group), and a group that received surgical treatment alone (control group). Three different types of comparisons were assessed, based on the treatment type. Five studies reported the comparison of open flap debridement (OFD) + APC versus OFD alone (Bajaj 2009 study evaluated the adjunctive effects of two different types of APCs in the same study, compared to OFD alone as the control group). Four studies reported the comparison of bone graft (BG) + APC versus BG alone, and only one study reported the comparison of guided tissue regeneration (GTR) + APC versus GTR alone. The results of these studies were separated for types of platelet concentrate data to facilitate subgroup meta-analysis for OFD + APC versus OFD comparison. However, it was impossible to undertake the subgroup analysis for the other two comparisons due to the lack of enough studies for sub-grouping. The risk of bias of all included studies is synthesized in Figure 2.

### 2.2. Meta-Analysis of Primary Outcomes

#### 2.2.1. Probing Pocket Depth (PPD)

##### APC + OFD vs. OFD Alone (Figure 3)

The forest plot of the included studies evaluating the change in PPD shows evidence of an advantage of using APC in adjunct to OFD (*p* < 0.001, mean difference 1.59, 95% CI 1.38, 1.80). The subgroup analysis is also favorable for both PRF (*p* < 0.001, mean difference 1.46, 95% CI 1.22, 1.70) and PRP (*p* < 0.001, mean difference 2.09, 95% CI 1.62, 2.55).

##### APC + BG vs. BG Alone (Figure 4)

The forest plot of the included studies evaluating the change in PPD in using APC in adjunct with BG favors the use of BG alone; however, the result is not statistically significant (*p* = 0.26, mean difference −0.08, 95% CI −0.22, 0.06).

#### 2.2.2. Vertical Clinical Attachment Level (VCAL)

##### APC + OFD vs. OFD Alone (Figure 5)

The forest plot of the included studies evaluating the change in VCAL shows evidence of an advantage of using APC in adjunct to OFD (*p* < 0.001, mean difference 1.24, 95% CI 1.08, 1.40). The subgroup analysis is also favorable for both PRF (*p* < 0.001, mean difference 1.18, 95% CI 1.01, 1.36) and PRP (*p* < 0.001, mean difference 1.58, 95% CI 1.17, 2.00).

##### APC + BG vs. BG Alone (Figure 6)

The forest plot of the included studies evaluating the change in VCAL in using APC in adjunct with BG is favorable; however, the result is not statistically significant (*p* = 0.62, mean difference 0.06, 95% CI −0.18, 0.30).

#### 2.2.3. Horizontal Clinical Attachment Level (HCAL)

##### APC + OFD vs. OFD Alone (Figure 7)

The forest plot of the included studies evaluating the change in HCAL shows evidence of an advantage of using APC in adjunct to OFD (*p* < 0.001, mean difference 1.01, 95% CI 0.89, 1.12). The subgroup analysis is also favorable for both PRF (*p* < 0.001, mean difference 0.93, 95% CI 0.80, 1.06) and PRP (*p* < 0.001, mean difference 1.50, 95% CI 1.18, 1.83).

##### APC + BG vs. BG Alone (Figure 8)

The forest plot of the included studies evaluating the change in HCAL shows evidence of an advantage of using APC in adjunct with BG (*p* < 0.001, mean difference 0.74, 95% CI 0.54, 0.94).

#### 2.2.4. Vertical Furcation Depth (VFD)

##### APC + OFD vs. OFD Alone (Figure 9)

The forest plot of the included studies evaluating the change in VFD shows evidence of an advantage of using APC in adjunct to OFD (*p* < 0.001, mean difference 1.60, 95% CI 1.53, 1.68). The subgroup analysis is also favorable for both PRF (*p* < 0.001, mean difference 1.65, 95% CI 1.57, 1.74) and PRP (*p* < 0.001, mean difference 1.38, 95% CI 1.21, 1.56).

##### APC + BG vs. BG Alone (Figure 10)

The forest plot of the included studies evaluating the change in VFD in using APC in adjunct with BG favors the use of BG alone; however, the result is not statistically significant (*p* = 0.90, mean difference −0.02, 95% CI −0.31, 0.27).

#### 2.2.5. Horizontal Furcation Depth (HFD)

##### APC + OFD vs. OFD Alone (Figure 11)

The forest plot of the included studies evaluating the change in HFD shows evidence of an advantage of using APC in adjunct to OFD (*p* < 0.001, mean difference 1.13, 95% CI 0.85,1.41). No subgroup analysis was carried out for this outcome due to the lack of enough studies.

##### APC + BG vs. BG Alone (Figure 12)

The forest plot of the included studies evaluating the change in VFD shows evidence of an advantage of using APC in adjunct with BG (*p* = 0.02, mean difference 0.17, 95% CI 0.02, 0.31).

## 3. Discussion

The use of platelet concentrates to promote periodontal regeneration has gained popularity in the last 10 years, as demonstrated by the increasing number of evidence-based randomized studies and systematic reviews [19,51,52]. A recent Cochrane systematic review [53] investigated the effect of APC for the surgical treatment of infrabony defects, reporting positive effects when APC is used in combination with OFD, OFD + BG, but not with GTR and enamel matrix derivative. The latter two treatments have a predictable and well-documented efficacy, and they are since long considered the gold standard for periodontal defects, so it can be difficult for any adjunctive therapy to further enhance the clinical outcomes. Evidence-based studies on the efficacy of APC for the regeneration therapy of furcation defects are relatively scarce as compared to infrabony defects. Our systematic review published in 2011 investigated the effects of APC on infrabony defects, gingival recessions and furcation defects but found only two studies on the latter topic, both using platelet-rich plasma [19]. The present study is the first comprehensive systematic review that was aimed at exploring and comparing the effect of various APCs for enhancing furcation treatment. It was designed according to a standard protocol, aimed at selecting only the best evidence studies, so as to provide the most reliable results. Only one of the ten included studies was judged at high risk of bias [50], while seven had a low risk, testifying to the good level of the evidence of this review. The results, derived from the analysis of different clinical outcome variables, suggested that the use of APC may be beneficial for improving the regeneration of furcation defects, when associated with OFD, in line with the above findings regarding infrabony defects. Further, it may be noted that APC in adjunct to OFD + BG also showed significant improvement in HCAL and HFL. Since only one RCT evaluated the adjunctive effect of APC when using GTR for grade II furcation defects, no meta-analysis was feasible. The results of this study, suggested that the adjunct of APC produced no significant advantage as compared to GTR alone, in line with previous findings for infrabony defects.

This review has some strengths and limitations. In recent years, there has been fierce competition among companies producing different types of platelet concentrates, all claiming that their product was superior to the others. This also introduced a number of different protocols for the preparation of APC. Indeed, very few studies comparing different types of APC have been performed in the periodontal field (as well as in other fields), so that it seems difficult to indicate if there is really a superiority of some APCs over the others for specific conditions. In the present review, we were able to perform a meta-analysis with subgroups, keeping separate different APC (PRP and PRF), only in the group considering OFD alone. The outcomes using different APC was very similar as can be seen in Figure 3, Figure 4, Figure 5 and Figure 6 This can be considered a strong point of the present review. However, the precise difference in effects between different APC cannot be estimated, due to a lack of direct comparisons. The same subgroup analysis could not be performed in the OFD + BG group, due to heterogeneity among studies in the type of APC used, and the insufficient number of studies using the same type of APC. Indeed, also when different studies use the same type of APC, this does not necessarily represent a warranty of homogeneity in the protocols. For example, over 20 different types of devices producing PRP are currently available on the market, and at least five different companies produce centrifuges for PRF [54]. A recent in vitro study compared the characteristics of PRF obtained using four different centrifuge systems [55]. This study found that, even though in all cases a leukocyte- and platelet-rich fibrin is obtained after centrifugation, the centrifuge characteristics and centrifugation protocols significantly impact the cell composition and distribution, the growth factors release pattern and the fibrin architecture of the final products. So, when PRF is used in different studies, one cannot be sure to refer to a product with the same features, unless the same centrifuge system is used. In spite of the above limitations, caused by lack of homogeneity in study protocols, it can be noted that all studies investigating the effect of APC as an adjunct to OFD alone, consistently reported a beneficial effect. The latter can be considered a strength point evidenced by this review.

In addition to the regenerative properties, platelet concentrates have also been demonstrated to carry further advantages in the postsurgical healing period. Evidence-based studies in different oral surgery procedures have reported that the adjunctive use of APC is associated with an improvement of patients’ quality of life and pain reduction in the post-surgical period [56,57] Unfortunately, such effects were not consistently addressed in the studies included in the present review.

Finally, though specific clinical studies have not been performed so far, there is consistent preclinical evidence that APCs have an antimicrobial effect against a number of species commonly found in the oral cavity, which suggest they may potentially represent a beneficial tool for the control of postsurgical infection [58,59]. Indications for future research: There is a huge demand for conducting more evidence-based comparative studies with wide sample size (among different APC and grafting materials and versus other biological agents), to investigate patients’ quality of life, to treat various grades of furcation, in order to verify the actual beneficial effects of use of APC as adjunct with wide variety of regenerative strategies.

## 4. Material and Methods

This systematic review and meta-analysis were carried out based on preferred reporting items for systematic reviews and meta-analysis (PRISMA) guidelines. The protocol of this systematic review was registered on the international prospective register of systematic reviews (PROSPERO) with registration number CRD42019100015.

### 4.1. Research Question

What is the effectiveness of autologous platelet concentrates used as an adjunct to different types of surgical techniques for the treatment of furcation defects, compared to the same surgical techniques alone?

### 4.2. Search Strategy

A systematic digitalized search was carried out in the following electronic databases: MEDLINE, SCOPUS, CENTRAL (Cochrane Central Register of Controlled Trials), and EMBASE, using a series of search terms combined with the Boolean Operators “AND”, “OR”, and “NOT”. The following search string was developed with the combination of relevant keywords: “(((Furcation Defects) OR Furcation Involvement)) AND (((((((Platelet Concentrates) OR Platelet-rich plasma) OR Platelet-rich fibrin) OR Growth factors) OR PRP) OR L-PRF) OR CGF)”. The last electronic search was carried out in October 2018. In addition, a hand search was performed in the following dental journals: British Dental Journal, British Journal of Oral and Maxillofacial Surgery, Clinical Implant Dentistry and Related Research, Clinical Oral Implants Research, Clinical Oral Investigations, European Journal of Oral Implantology, European Journal of Oral Sciences, Implant Dentistry, International Journal of Oral and Maxillofacial Implants, International Journal of Oral and Maxillofacial Surgery, International Journal of Periodontics and Restorative Dentistry, Journal of Clinical Periodontology, Journal of Dental Research, Journal of Dentistry, Journal of Implantology, Journal of Maxillofacial and Oral Surgery, Journal of Oral and Maxillofacial Surgery, Journal of Periodontal Research, Journal of Periodontology, and Oral Surgery, Oral Medicine, Oral Pathology, Oral Radiology. The reference citations of the eligible studies and other systematic reviews were also searched for possible additional eligible studies. Finally, the online trial registries were also searched for any ongoing studies: US National Institutes of Health Ongoing Trials Register ClinicalTrials.gov (clinicaltrials.gov; searched 20 October 2018); World Health Organization International Clinical Trials Registry Platform (apps.who.int/trialsearch; searched 20 October 2018). No language restrictions were applied.

### 4.3. Inclusion Criteria

The criteria for the articles to be included in this present systematic review were as follows: Randomized clinical trials (RCT), either of a parallel group or of a split-mouth design;Presence of at least one experimental group in which APCs were clinically applied as an adjunct to surgical procedures alone or in combination with bone grafting materials or GTR procedures for the therapy of furcation defects;Presence of an appropriate control group, in which the same therapeutic procedures as those employed in at least one experimental group were clinically applied for the treatment of furcation defects, without the adjunctive effect of APCs;Patients included in the RCT should present with maxillary/mandibular Grade 2 or 3 furcation defects;Patients included in the RCT should have no systemic diseases nor taking medications that could potentially influence the outcome of periodontal therapy;The follow-up period had to be at least 6 months.

### 4.4. Selection of Studies

Following the electronic search in all the respective databases, the records were imported into EndNote 13 software (EndNoteX3; Thomas Reuters, New York, NY, USA) and the duplicates were sorted to be removed from the pool of titles. A total of 153 titles and abstracts (if available) were independently screened by two reviewers (MDF, SP) to exclude all articles clearly not meeting the inclusion criteria. Of all the eligible articles, full texts were obtained and were thoroughly assessed. Only articles fulfilling the inclusion criteria were considered. In cases of disagreement between the two reviewers, a third reviewer (LF) was consulted. Detailed reasons were stated for all excluded studies.

### 4.5. Data Extraction and Management

The relevant data of the included studies were extracted using an Excel spreadsheet (Microsoft, Radmond, WA, USA). Data were independently extracted by two review authors (MDF, FG) and recorded on predetermined spreadsheets. In case of missing or unclear information, the authors of the included studies were contacted by email for providing clarification or missing information.

The following data were recorded for each included report:Patients’ demographic characteristicsStudy design and sample sizeType of platelet concentrate used (PRP, PRF, PRGF, CGF)Follow up durationsource of funding and study settingOutcome variables, relative to baseline and post-operative defect characteristics (probing pocket depth (PPD), horizontal and vertical clinical attachment loss (HCAL, VCAL), horizontal and vertical furcation depth (HFD, VFD)

### 4.6. Risk of Bias Assessment

Risk of Bias was assessed by two independent reviewers (ACD, AS) for all the included clinical trials and the discrepancies were resolved by discussion and in consent with a third reviewer (MK). The domains of the study were graded under high, unclear or low risk, based on the following categories: Selection bias (random sequence generation and allocation concealment), performance bias (blinding), detection bias (assessor blinding), attrition bias (incomplete outcome data), reporting bias (selective reporting), sample size calculation and number of surgeons involved. Based on the domains, the studies categorized as low risk of bias if all domains were at low risk; high risk of bias if two or more domains were at high risk; or medium risk of bias if one domain were at unclear or high risk.

### 4.7. Data Synthesis

Data of the various outcomes were extracted from each included study. Parallel group and split-mouth studies were combined in the meta-analysis of treatment effects. For all the outcomes, mean differences and 95% confidence interval (CI) were used to summarize the results for each included study. The meta-analysis was performed using Review Manager 5.3 software (RevMan 5.3, Version 5.3.5 Copenhagen: The Nordic Cochrane Centre, The Cochrane Collaboration, 2014) using the fixed or random effects models, as appropriate. Fixed effects meta-analysis was used when the heterogeneity was small (*i*^2^ < 60%, *p* > 0.05). When the heterogeneity was large (*i*^2^ > 60%, *p* < 0.05), a random-effects model analysis was undertaken.

## 5. Conclusions

In conclusion, the evidence available in the literature for the beneficial effects of platelet concentrates in periodontal furcation defects has been increasing in recent years. Platelet-rich plasma and platelet-rich fibrin may be advantageously used as an adjunct to open flap debridement alone and additional grafting procedures, while there is no evidence of an effect of APC when used in combination with GTR, for the treatment of furcation defects.

## Figures and Tables

**Figure 1 ijms-20-01347-f001:**
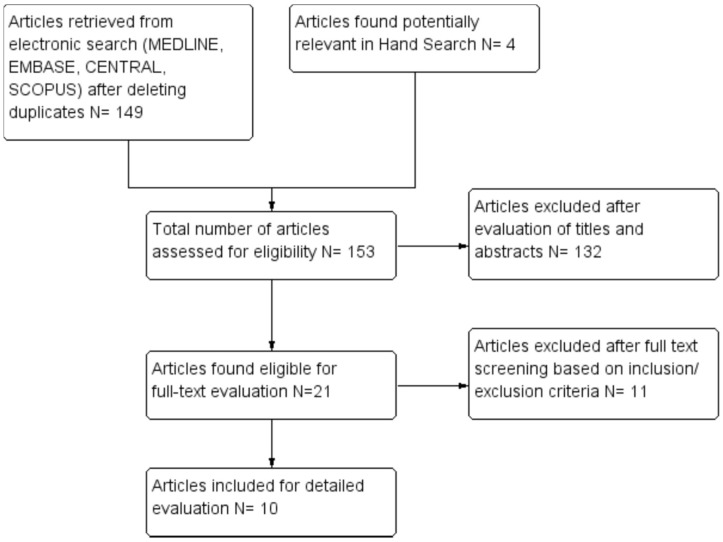
Flow chart depicting study selection process.

**Figure 2 ijms-20-01347-f002:**
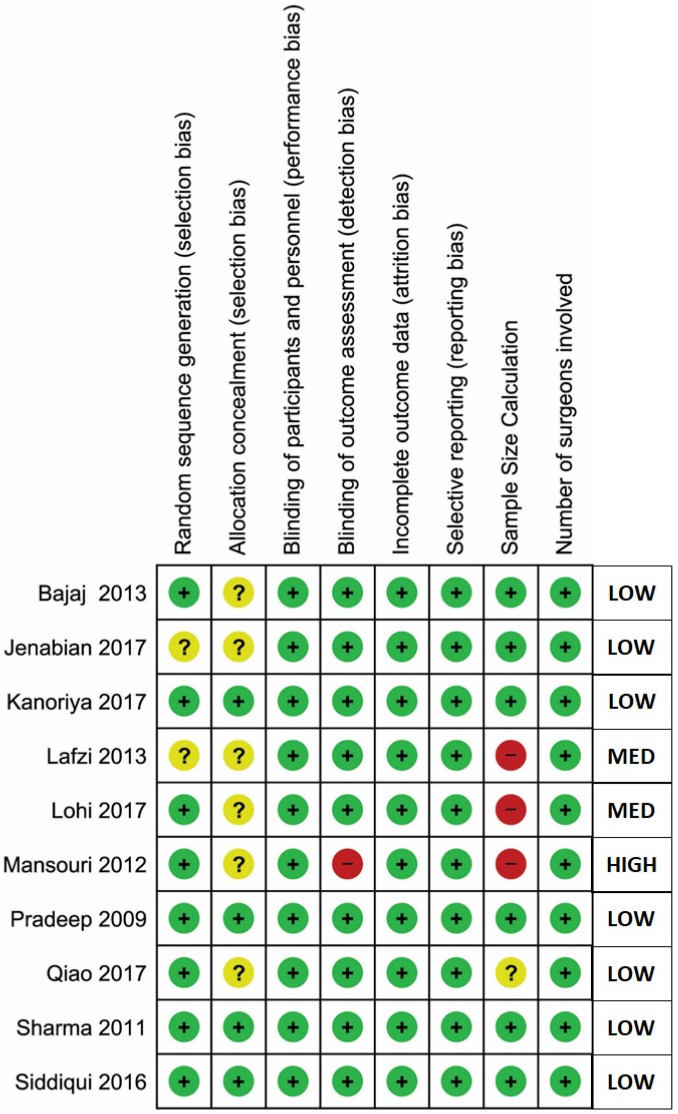
Risk of bias (RoB) assessment.

**Figure 3 ijms-20-01347-f003:**
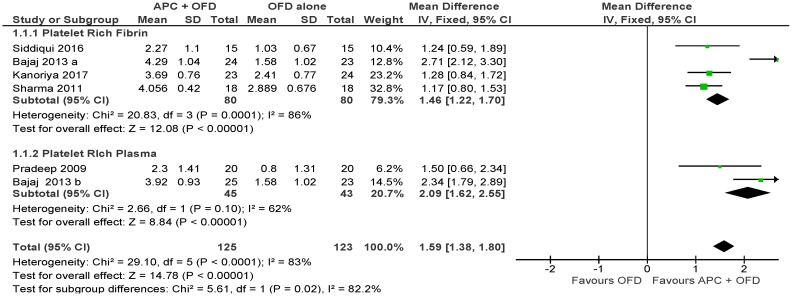
Forest plot showing the effect on probing pocket depth for comparison of APC + OFD versus OFD alone at end of all-follow-up (6–12 m).

**Figure 4 ijms-20-01347-f004:**
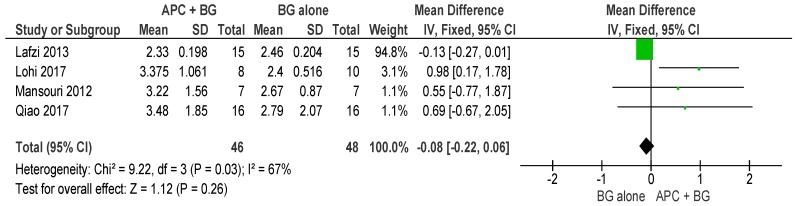
Forest plot showing the effect on probing pocket depth for comparison of APC + BG versus BG alone at end of all-follow-up (6–12 m).

**Figure 5 ijms-20-01347-f005:**
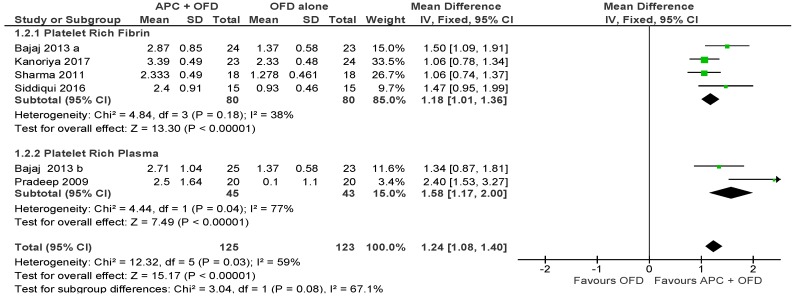
Forest plot showing the effect on vertical clinical attachment level for comparison of APC + OFD versus OFD alone at end of all-follow-up (6–12 m).

**Figure 6 ijms-20-01347-f006:**
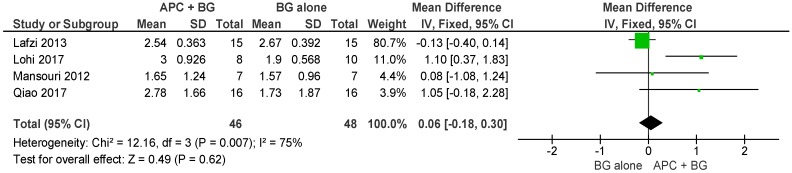
Forest plot showing the effect on vertical clinical attachment level for comparison of APC + BG versus BG alone at end of all-follow-up (6–12 m).

**Figure 7 ijms-20-01347-f007:**
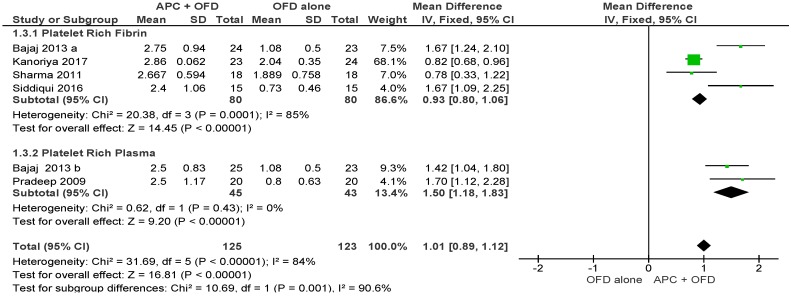
Forest plot showing the effect on horizontal clinical attachment level for comparison of APC + OFD versus OFD alone at end of all-follow-up (6–12 m).

**Figure 8 ijms-20-01347-f008:**
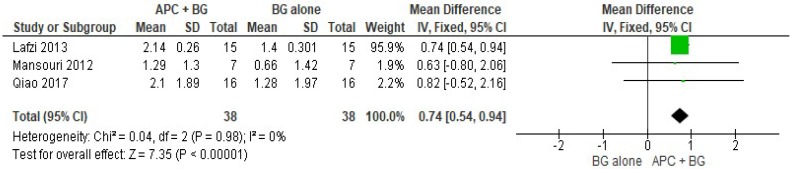
Forest plot showing the effect on horizontal clinical attachment level for comparison of APC + BG versus BG alone at end of all-follow-up (6–12 m).

**Figure 9 ijms-20-01347-f009:**
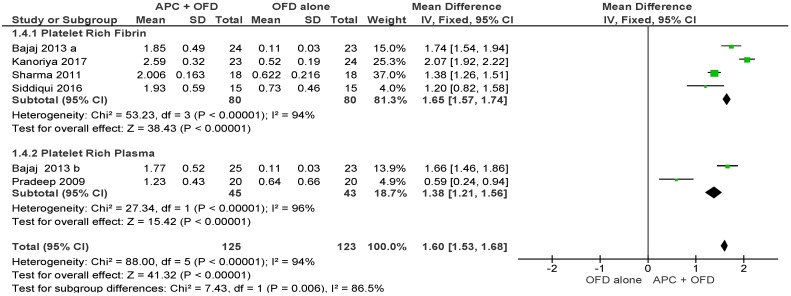
Forest plot showing the effect on vertical furcation depth for comparison of APC + OFD versus OFD alone at end of all-follow-up (6–12 m).

**Figure 10 ijms-20-01347-f010:**
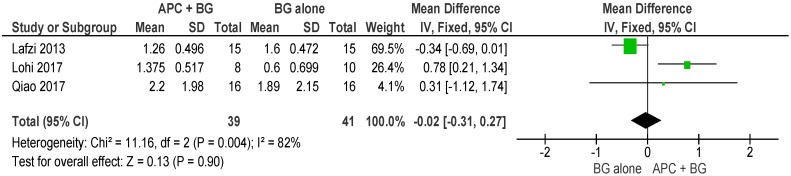
Forest plot showing the effect on vertical furcation depth for comparison of APC + BG versus BG alone at end of all-follow-up (6–12 m).

**Figure 11 ijms-20-01347-f011:**
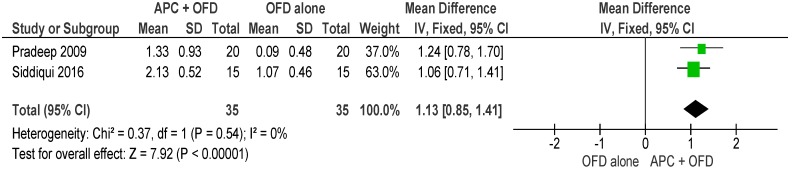
Forest plot showing the effect on horizontal furcation length for comparison of APC + OFD versus OFD alone at end of all-follow-up (6–12 m).

**Figure 12 ijms-20-01347-f012:**
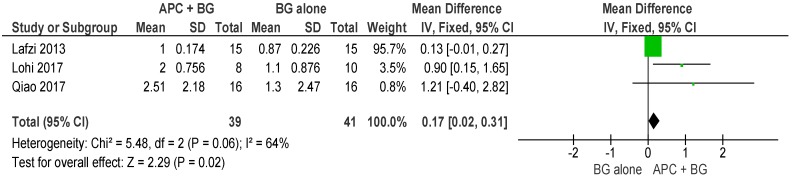
Forest plot showing the effect on horizontal furcation length for comparison of APC + BG versus BG alone at end of all-follow-up (6–12 m).

**Table 1 ijms-20-01347-t001:** Various proposed classification systems of furcation involvement.

Sl	Author	Year	Classification System
**Horizontal Component**			
1	Goldman, H.M [20]	1958	Grade I: Incipient lesion; Grade II: Cul-de-sac lesion; Grade III: Through-and-through lesion
2	Staffileno, H.J. [21]	1969	Class I: Furcations with a soft tissue lesion extending to furcal level but with a minor degree of osseous destruction; Class II: Furcations with a soft tissue lesion and a variable degree of osseous destruction but not a through-and-through communication through the furca; Class III: Furcations with osseous destruction with through-and-through communication
3	Glickman, I. [4]	1972	Grade I: Incipient lesion. Suprabony pocket and slight bone loss in the furcation area. Grade II: Loss of interradicular bone and pocket formation but a portion of the alveolar bone and periodontal ligament remain intact. Grade III: Through-and-through lesion. Grade IV: Through-and-through lesion with gingival recession, leading to a clearly visible furcation area.
4	Hamp, S.E. et al. [22]	1975	Degree I: Horizontal attachment loss < 3 mm; Degree II: Horizontal attachment loss > 3 mm not encompassing the width of the furcation area; Degree III: Horizontal through-and-through destruction of the periodontal tissue in the furcation area.
5	Ramjford, S.P. et al. [23]	1979	Class I: Tissue destruction < 2 mm (1/3 of tooth width) into the furcation; Class II: Tissue destruction > 2 mm (>1/3 of tooth width); Class III: Through-and-through involvement.
6	Richietti, P.A. [24]	1982	Class I: 1 mm of horizontal invasion; Class Ia. 1–2 mm of horizontal invasion; Class II: 2–4 mm of horizontal invasion; Class IIa. 4–6 mm of horizontal invasion; Class III: >6 mm of horizontal invasion.
7	Grant, D.A. et al. [25]	1988	Class I: Involvement of the flute only; Class II: Involvement partially under the roof; Class III: Through-and-through loss.
8	Goldman, H.M. et al. [26]	1988	Degree I: Involves furcation entrance; Degree II: Involvement extends under the roof of furcation; Degree III: Through-and-through involvement.
9	Basaraba, N. [27]	1990	Class I: Initial furcation involvement; Class II: Partial furcation involvement; Class III: Communicating furcation involvement.
10	Nevins, M. et al. [28]	1998	Class I: Incipient or early loss of attachment; Class II: A deeper invasion and loss of attachment that does not extend to a complete invasion; Class III: Complete loss of periodontium extending from buccal to lingual surface. Diagnosed radiographically and clinically
11	Walter, C.et al [29]	2009	Modification of the Hamp et al. classification. Degree I: Horizontal attachment loss < 1/3 of the width of the tooth; Degree II: Horizontal loss of support > 3 mm, < 6 mm; Degree II–III: Horizontal loss of support > 6 mm, but not extending completely. Degree III: Horizontal through-and-through destruction.
12	Carnevale, G. et al. [30]	2012	Degree I: Horizontal attachment loss < 1/3; Degree II: Horizontal attachment loss > 1/3; Degree III: Horizontal through-and-through destruction.
**Vertical Component**			
1	Tal, H. et al. [31]	1982	Furcal rating 1: Depth of the furcation is 0 mm; Furcal rating 2: Depth of the furcation is 1–2 mm; Furcal rating 3: Depth of the furcation is 3 mm; Furcal rating 4: Depth of the furcation is 4 mm or more.
2	Eskow, R.N. et al. [32]	1984	Furcation involvement grade 1 is classified as: Subclass A: Vertical destruction > 1/3; Subclass B: Vertical destruction of 2/3; Subclass C: Vertical destruction beyond the apical third of interradicular height.
3	Tarnow, D. et al. [33]	1985	For each class of horizontal classification (I–III), a subclass based on the vertical bone resorption was added: Subclass A: 0–3 mm; Subclass B: 4–6 mm; Subclass C: >7 mm.
**Horizontal & Vertical Component (Combined)**			
1	Easley, J.R. et al. [34]	1969	Class I: Incipient involvement, but there is no horizontal component to the furca; Class II: Type 1. Horizontal attachment loss into the furcation; Type 2. Vertical attachment loss into the furcation; Class III: Through-and-through attachment loss into the furcation; Type 1. Horizontal attachment loss into the furcation; Type 2. Vertical attachment loss into the furcation.
2	Fedi, P.F. [35]	1985	Glickman + Hamp classifications: Grades are the same as Glickman’s classification (I–IV); Grade II is subdivided into degrees I and II; Degree I. Vertical bone loss 1–3 mm; Degree II. Vertical bone loss > 3 mm, but not communicate through-and-through.
3	Rosemberg, M.M. [36]	1986	Horizontal: Degree I: Probing < 4 mm; Degree II: Probing > 4 mm; Degree III: Two or three furcations classified as degree II are found. Vertical: Shallow: Slight lateral extension of an interradicular defect, from the center of the trifurcation in a horizontal direction; Deep: Internal furcation involvement but not penetrating the adjacent furcation.
4	Hou, G.L. et al. [37]	1998	Classification based on root trunk length and horizontal and vertical bone loss. Types of root trunk: Type A: Furcation involving a cervical third of root length; Type B: Furcation involving a cervical third and two thirds of root length; Type C: Furcation involving a cervical two thirds of root length. Classes of furcation: Class I: Horizontal loss of 3 mm; Class II: Horizontal loss > 3 mm; Class III: Horizontal “through-and-through” loss. Subclasses by radiographic assessment of the periapical view: Sub-class ‘a’. Suprabony defect; Sub-class ‘b’. Infrabony defect.

**Table 2 ijms-20-01347-t002:** List of excluded studies.

Study & Year	Reason for Exclusion
Mehta et al. 2018 [38]	Use of Collagen Membrane along with DFDBA in control group
Wanikar et al. 2018 [39]	Both Control and Experimental group use PRF
Kaur et al. 2018 [40]	Both Control and Experimental group use PRF
Sharma et al. 2017 [41]	Both Control and Experimental group use PRF
Asimuddin et al. 2017 [42]	Comparison between use of PRF and Allograft + GTR
Salaria et al. 2016 [43]	Case Report
Biswas et al. 2016 [44]	Comparison between PRF and Bioactive Glass.
Pradeep et al. 2016 [45]	Both Control and Experimental group uses PRF
Sandhu et al. 2015 [46]	Case Report
Mellonig et al. 2009 [47]	Histological assessment
Lekovic et al. 2003 [48]	Comparison of PRP/BPBM/GTR versus OFD alone

PRF—platelet-rich fibrin, PRP—platelet-rich plasma, DFDBA—de-mineralized freeze-dried allograft, BPBM—bovine porous bone mineral, GTR—guided tissue regeneration.

**Table 3 ijms-20-01347-t003:** Characteristics of included studies.

Study & Year	Study Design	RCT Type	Treatment Comparison	N. Defects Test/Control	Age Range	Gender M/F	Follow Up
Kanoriya et al. 2017 [13]	RCT	Parallel	OFD vs. OFD + PRF	26/26	30–50 (38)	36/36	9 m
Siddiqui et al. 2016 [49]	RCT	Parallel	OFD vs. OFD + PRF	17/17	30–50	24/7	6 m
Bajaj et al. 2013 [11]	RCT	Parallel	OFD vs. OFD + PRP OFD vs. OFD + PRF	27/27 27/27	39.4	22/20	9 m
Sharma et al. 2011 [17]	RCT	Split Mouth	OFD vs. OFD + PRF	18/18	34.2	10/8	9 m
Pradeep et al. 2009 [6]	RCT	Split Mouth	OFD vs. OFD + PRP	20/20	42.8	10/10	6 m
Lohi et al. 2017 [14]	RCT	Parallel	BCCG + PRF vs. BCCG alone	10/10	25–65 (43.05 + 10.73)	12/4	6 m
Lafzi et al. 2013 [15]	RCT	Parallel	ABG + PRGF vs. ABG alone	15/15	NR	NR	6 m
Mansouri et al. 2012 [50]	RCT	Split Mouth	BPBM + PRGF vs. BPBM alone	7/7	44.7 + 11.2	4/3	6 m
Qiao et al. 2017 [16]	RCT	Parallel	BG + CGF vs. CGF alone	15/16	NR	15/5	12 m
Jenabian et al. 2017 [12]	RCT	Split Mouth	GTR + PRGF vs. GTR alone	8/8	NR	NR	6 m

RCT—randomized clinical trial, OFD—open flap debridement, PRF—platelet-rich fibrin, PRP—platelet-rich plasma, BCCG—bioactive ceramic composite granules, ABG—autogenous bone graft, BG—bone graft, GTR—guided tissue regeneration.

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
