# Peer review of "Autologous Platelet Concentrates in Treatment of Furcation Defects—A Systematic Review and Meta-Analysis"

_ijms, 2019, doi:10.3390/ijms20061347_

Round 1

Reviewer 1 Report

It's a good job, I only suggest the following:

1. improve figure 1

2. Table 1, divide it in two.

Author Response

Reviewer 1:

It's a good job, I only suggest the following:

1. improve figure 1

2. Table 1, divide it in two.

Response from authors:

Thank you for your esteemed suggestion.

1.       Figure 1 is the digitalized output of the figure by Review Manager 5.3 software (RevMan 5.3, Version 5.3.5 Copenhagen: The Nordic Cochrane Centre, The Cochrane Collaboration, 2014.)

2.       Table 1 is now divided into two tables (Table 1a, Table 1b)

Reviewer 2 Report

The manuscript presents a systematic review together with meta-analysis on the use of autologous platelet concentrates in the treatment of furcation defects.

This is an interesting review with a good level of evidence that allow the comparison between the effect of various APCs for enhancing furcation treatment. The manuscript is well-written with appropriate language and a clear research question. The methods are well described, and PRISMA guidelines were correctly followed. The discussion does not over-interpret the results obtained and relevant and current literature is well considered.

There are some points that need to be addressed:

1)      Abstract. Page 1, line 31. Please, include here the abbreviation GTR after guided tissue regeneration since it appears later alone in line 34.

2)      Page 2, line 52. The sentence requires revision.

3)      Page 12, line 292. Please check the spelling in ´pocket`.

4)      Would it be possible to increase the size of forest plots to make them easier to read? Please unify the format (font/size) in all the forest plots.

5)      Discussion. Page 14, lines 329-332. The authors mention that the use of APC may be beneficial for improving the regeneration of furcation defects when associated with OFD+BG. However, the association OFG+BG is not shown in the results presented. Please clarify this point.

6)      Please mention the limitations of the study.

Author Response

Reviewer 2:

The manuscript presents a systematic review together with meta-analysis on the use of autologous platelet concentrates in the treatment of furcation defects.

This is an interesting review with a good level of evidence that allow the comparison between the effect of various APCs for enhancing furcation treatment. The manuscript is well-written with appropriate language and a clear research question. The methods are well described, and PRISMA guidelines were correctly followed. The discussion does not over-interpret the results obtained and relevant and current literature is well considered.

There are some points that need to be addressed:

1)          Abstract. Page 1, line 31. Please, include here the abbreviation GTR after guided tissue regeneration since it appears later alone in line 34.

Response from authors: Thank you for your suggestion. The abbreviation GTR has been included right next to guided tissue regeneration, within paranthesis.

2)          Page 2, line 52. The sentence requires revision.

Response from authors: Thank you for your suggestion. The sentence has been revised as follows “Non-surgical strategies such as scaling and root planing, furcation-plasty etc. are employed to treat the furcations with Grade I intial involvement which restores the gingival health.”

3)          Page 12, line 292. Please check the spelling in ´pocket`.

Response from authors: Thank you for your detailed observation. The correction has been made.

4)          Would it be possible to increase the size of forest plots to make them easier to read? Please unify the format (font/size) in all the forest plots.

Response from authors: Thank you for your suggestion. The size of the images pertaining to the forest plots has been increased and unified.

5)          Discussion. Page 14, lines 329-332. The authors mention that the use of APC may be beneficial for improving the regeneration of furcation defects when associated with OFD+BG. However, the association OFG+BG is not shown in the results presented. Please clarify this point.

Response from authors: Thank you for your esteemed observation. The line would now be read as follows “The results, deriving from the analysis of different clinical outcome variables, suggested that the use of APC may be beneficial for improving the regeneration of furcation defects, when associated with OFD, in line with the above findings regarding infrabony defects.Further, it may be noted that APC in adjunct to OFD+BG also showed significant improvement in HCAL and HFL.” Thank you.

6)          Please mention the limitations of the study.

Response from authors: Thank you for your suggestion. The limitations of the present review have been better highlighted in the discussion.

Reviewer 3 Report

In this review, the authors evaluated the adjunctive effect of autologous platelet concentrates in treatment of 87 furcation defects both qualitatively and quantitatively, in terms of scientific quality of the clinical 88 trials and regeneration parameters assessment.

The paper is well structured and written and I think that this review could be of great interest for the readers of International Journal of Molecular Science.

For this reasons I recommended acceptance the article.

Author Response

Reviewer 3:

In this review, the authors evaluated the adjunctive effect of autologous platelet concentrates in treatment of 87 furcation defects both qualitatively and quantitatively, in terms of scientific quality of the clinical trials and regeneration parameters assessment.

The paper is well structured and written and I think that this review could be of great interest for the readers of International Journal of Molecular Science.

For this reasons I recommended acceptance the article.

Response from authors: Thank you for recommendation.